# Maintenance of Skeletal Muscle to Counteract Sarcopenia in Patients with Advanced Chronic Kidney Disease and Especially Those Undergoing Hemodialysis

**DOI:** 10.3390/nu13051538

**Published:** 2021-05-02

**Authors:** Katsuhito Mori

**Affiliations:** Department of Nephrology, Osaka City University Graduate School of Medicine 1-4-3, Asahi-Machi, Abeno-ku, Osaka 545-8585, Japan; ktmori@med.osaka-cu.ac.jp; Tel.: +81-6-6645-3806; Fax: +81-6-6645-3808

**Keywords:** skeletal muscle, sarcopenia, hemodialysis, aging, chronic kidney disease, diabetes

## Abstract

Life extension in modern society has introduced new concepts regarding such disorders as frailty and sarcopenia, which has been recognized in various studies. At the same time, cutting-edge technology methods, e.g., renal replacement therapy for conditions such as hemodialysis (HD), have made it possible to protect patients from advanced lethal chronic kidney disease (CKD). Loss of muscle and fat mass, termed protein energy wasting (PEW), has been recognized as prognostic factor and, along with the increasing rate of HD introduction in elderly individuals in Japan, appropriate countermeasures are necessary. Although their origins differ, frailty, sarcopenia, and PEW share common components, among which skeletal muscle plays a central role in their etiologies. The nearest concept may be sarcopenia, for which diagnosis techniques have recently been reported. The focus of this review is on maintenance of skeletal muscle against aging and CKD/HD, based on muscle physiology and pathology. Clinically relevant and topical factors related to muscle wasting including sarcopenia, such as vitamin D, myostatin, insulin (related to diabetes), insulin-like growth factor I, mitochondria, and physical inactivity, are discussed. Findings presented thus far indicate that in addition to modulation of the aforementioned factors, exercise combined with nutritional supplementation may be a useful approach to overcome muscle wasting and sarcopenia in elderly patients undergoing HD treatments.

## 1. Introduction

Most societies, especially those in developed countries, have shown increased longevity over the past few generations and interest has now shifted to how to extend healthy life expectancy. Aging is profoundly associated with changes related to human organs and tissues, and recently ‘frailty’ has become recognized as a key term related to age-related decline [1]. A sedentary lifestyle, commonly seen in modern society settings, also accelerates deterioration of motor functions. Skeletal muscle, which constitutes the largest type of tissue mass and accounts for 40–45% of total body weight [2], has a core role in maintenance of a healthy life. Its functional failure leads to physical impairment, resulting in poor outcomes, especially in elderly individuals. Thus, much attention has been given to ‘sarcopenia’, which is generally defined as loss of skeletal muscle mass and function [3].

In addition to aging, other chronic disorders are known to exacerbate frailty and/or sarcopenia, with advanced chronic kidney disease (CKD), especially end-stage kidney disease including hemodialysis (HD), a representative condition [4]. ‘Protein-energy wasting (PEW)’ is characterized by adverse changes in nutrition and body composition in advanced CKD/HD patients [5]. Historically, frailty and sarcopenia have been considered to originate from aging-related derangement, and PEW has been proposed to express the wasting that occurs in association with kidney dysfunction [6]. As a result, there is considerable overlapping among frailty, sarcopenia, and PEW in elderly patients with advanced CKD/HD. 

The aim of this report is not to provide a systematic review of sarcopenia in advanced CKD/HD cases, but rather to examine the wide range of related fields in an easily understood manner in order to facilitate research regarding skeletal muscle maintenance and healthy life expectancy. Specifically, the author would like to focus on the decline and dysfunction of skeletal muscle, along with countermeasures from the viewpoint of aging and advanced CKD/HD as common key components of PEW, sarcopenia, and frailty (Figure 1).

## 2. Conceptual Overlapping among Frailty, Sarcopenia, and PEW

The concept of frailty is considered acceptable to describe the condition of an individual. Generally, it is used to explain the state resulting from an age-related decrease in physiological reserve and increase in vulnerability to stressors, resulting in disability, hospitalization, institutionalization, and finally death [4,7]. Fried et al. defined frailty as a clinical syndrome in which three or more of the following abnormalities were combined; unintentional weight loss, self-reported exhaustion, weakness (grip strength), slow walking speed, and low physical activity [1]. In contrast to that phenotype model, Rockwood et al. proposed a frailty index based on accumulation of such deficits as age-associated diseases, non-specific vulnerability, and disabilities (accumulated deficit model) [8]. Nevertheless, no universal standard for diagnosis of frailty has been established. 

The term ‘sarcopenia’ was first proposed by Irwin Rosenberg to describe loss of muscle mass (i.e., *sarx* meaning flesh and *penia* loss in Greek) [9]. However, the concept of sarcopenia has changed over time and later included related dysfunctions, such as loss of muscle strength. The European Working Group on Sarcopenia in Older People (EWGSOP) provided a definition along with diagnostic criteria [10], an objective assessment based on measurements of gait speed, grip strength, and muscle mass that has greatly contributed to progress in research of sarcopenia. Recently, a revised consensus (EWGSOP2) was released to promote early detection and treatment of affected patients [3]. 

In addition to age-related frailty and sarcopenia, the presence of advanced CKD is independently associated with malnutrition and inflammation. Previously, these conditions were expressed by various terms such as uremic malnutrition [11], uremic cachexia [12], protein-energy malnutrition [13], and malnutrition-inflammation complex syndrome [14] as well as others. Since CKD is commonly associated with atherosclerosis, the term, malnutrition-inflammation atherosclerosis (MIA) syndrome, was also proposed [15]. However, malnutrition includes both under- and overnutrition, and some CKD patients are underweight even with adequate intake. To avoid confusion, the International Society of Renal Nutrition and Metabolism (ISRNM) proposed the nomenclature protein-energy wasting (PEW) for loss of muscle and fat tissues (wasting), or the presence of malnutrition and/or inflammation [5]. For diagnosis of PEW, four categories including biochemical criteria; low body weight, reduced total body fat, or weight loss; decreased muscle mass; and low protein or energy intake, are evaluated [5]. Although useful, an evaluation of chronic inflammation is not necessary for determining a diagnosis. Among those categories, decreased muscle mass is considered to be the most valid criterion for PEW determination [5]. 

Frailty, sarcopenia, and PEW share common components in elderly patients undergoing HD (Figure 1). Following, this report will mainly focus on sarcopenia in HD patients, including consideration of the physiology and pathology of skeletal muscle. 

## 3. Diagnosis of Sarcopenia in Asians Including Japanese

As noted above, the definition of sarcopenia and criteria used for diagnosis presented by EWGSOP were epoch-making. However, many problems remain to be solved. To develop appropriate measures against sarcopenia, it is necessary to obtain an accurate understanding of its prevalence, as various reports have shown a wide range from 4 to 63% of CKD patients [16], with those findings largely dependent on the methods, cut-off values, and criteria employed. Furthermore, it is also important to consider age- and/or CKD- related muscle histological modifications such as myosteatosis and myofibrosis [17,18], which are described later. Clinically, evaluation of muscle mass is more problematic than that of muscle strength, which is usually assessed based on handgrip strength [3,10,19,20]. For measuring muscle quantity, various methods, including dual-energy X-ray absorptiometry (DXA), bioimpedance analysis (BIA), mid-arm muscle circumference (MAC), and sum of skinfold thickness (SKF), are available. DXA is useful for evaluating body composition [21], while on the other hand the precision of BIA, MAC, and SKF remains controversial, though those are non-invasive and inexpensive [19]. In fact, the prevalence of low muscle mass evaluated by different methods (DXA, BIA, MAC, SKF) showed a wide range of variation from 4.0 to 73.5% [22]. Another problem is related to the different normalizing methods used for muscle mass. For example, muscle mass can be indexed by height squared, percentage of body weight, body surface area, and BMI. Those four different normalization methods were compared in 645 patients undergoing HD [23], with the presence of low muscle mass defined as two standard deviation (SD) below the normal mean of young adults. Intriguingly, the prevalence rate of low muscle mass ranged from 8.1 to 32.4%, even when muscle mass was the same [23]. Thus, a standard definition for sarcopenia is necessary for accurate evaluation of affected individuals. Another problem may be that the adaptation of EWGSOP criteria for Asian patients can be problematic because of anthropometric as well as cultural and/or life-style-related differences as compared with Europeans. As a result, the Asian Working Group on Sarcopenia (AWGS) established criteria for Asian populations in 2014 [19]. 

Using the AWGS criteria, we carefully evaluated sarcopenia in 308 Japanese patients receiving HD, and determined muscle strength using handgrip strength and muscle mass measurements by DXA. That study reported a sarcopenia prevalence rate (40%) in Japanese HD patients based on the AWGS criteria 2014 [24]. More importantly, the results showed that the presence of sarcopenia was a significant predictor of all-cause mortality in older patients. Thus far, studies regarding sarcopenia according to those criteria are limited, though reports showing prevalence rates in CKD [25] and peritoneal dialysis [26], and kidney transplant recipients [27] of 25.0%, 10.9%, and 11.8%, respectively, have been presented. For interpreting results, age distribution should also be considered. Very recently, the AWGS consensus has been revised and the current version is AWGS 2019 [28]. More careful methods for diagnosis and appropriate intervention are expected. 

## 4. Which Component, Muscle Mass or Muscle Strength, Is Critical for Prognosis?

Although loss of muscle mass and reduced muscle strength are both related to aging, they do not always occur in parallel. In longitudinal observational studies of older healthy adults, a decline in muscle strength appeared to precede muscle loss [29,30], thus suggesting a dissociation between them. Such a dissociation between was observed in 111 HD patients [31], in whom the prevalence of low muscle strength and low muscle mass based on cut-off values in EWGSOP criteria was 88.3% and 33.3%, respectively. Although 31.5% of those was finally diagnosed as sarcopenia, that was largely dependent on low muscle mass, since the majority of the patients (88.3%) showed low muscle strength. Previously, we focused on muscle quality in Japanese patients on HD [32]. Some in the population, such as those affected by diabetes, showed a lower serum creatinine level, possibly reflecting reduced muscle mass, thus we adopted handgrip strength per unit of arm muscle mass determined using DXA to show muscle quality [32]. With this approach, muscle quality was demonstrated to be a predictor of high mortality in Japanese patients undergoing HD independent of age, serum albumin, and presence of diabetes [33]. A later study also showed that handgrip strength was a predictor of all-cause mortality in patients on HD, though muscle mass was not evaluated [34]. Similarly, in an observational retrospective cohort study, the usefulness of handgrip strength as a survival predictor was confirmed in patients receiving HD and peritoneal dialysis [35]. A simple question to determine is which component, muscle mass or muscle strength, can better predict mortality in HD patients. Along that line, both muscle strength and muscle mass were measured at the baseline in incident dialysis patients, then their impact on mortality was examined [36]. The results demonstrated that patients with low muscle strength but not those with low muscle mass were at increased risk of mortality. Similarly, an independent work revealed that muscle strength was a more relevant predictor of survival in patients on HD as compared with low muscle mass [23]. As well as handgrip strength, decreased muscle strength in the lower extremities was also found to be strongly associated with increased mortality in HD patients [37]. Although evaluation of muscle mass is necessary for assessing nutritional status (wasting) in advanced CKD/HD cases, handgrip strength measurement should be incorporated to evaluate physical performance in clinical practice, as it is easy to perform and inexpensive [16]. 

## 5. Physiology of Skeletal Muscle

Recently, detailed studies have provided advanced knowledge regarding the association of CKD and/or aging-related factors with pathological and functional changes of skeletal muscle. At the same time, the abundant information available can be confusing for non-experts. Therefore, relevant summaries showing the basic morphology and physiology of skeletal muscle for understanding pathophysiology are important. The availability of such materials can help with deep communication among specialists in each field.

### 5.1. Development and Regeneration of Skeletal Muscle

Skeletal muscle consists of muscle fibers, or myofibers, which function as a syncytium originating from the fusion of myoblasts [2]. During development, myoblasts, mono-nucleated muscle precursor cells, fuse together to generate nascent myotubes that exhibit central nucleation. The nuclei are located in the central portion of nascent myotubes. When muscle fibers become mature, they migrate to the periphery of the myofibers [2]. Satellite cells, stem cells of skeletal muscle, are located between the basal lamina and plasma membrane of muscle fibers, where they proliferate and differentiate into myoblasts in response to diverse stimuli, such as injury, exercise, stretching, and denervation [38]. Upon stimulation, some satellite cells differentiate into myoblasts and subsequently fuse with exiting fibers (regeneration). During this process, another small proportion returns to quiescence to form a new pool of myoblasts (self-renewal) (Figure 2) [39]. Although the mechanisms of self-renewal are poorly understood, one of the postulated models shows that asymmetric cell division can produce two types of daughter cells, those committed as myogenic precursor cells for regeneration and other uncommitted pluripotent cells involved in self-renewal [38]. 

As satellite cell markers, paired domain transcription factors such as Pax7 and Pax3, and the myogenic factor 5 (Myf5) are well known. Among those, Pax7 is expressed in all quiescent and proliferating satellite cells in various species including humans [40]. During development, myogenic regulatory factors are required for myoblast commitment and differentiation, and the primary factors Myf5 and MyoD are necessary for determination of myoblasts. Subsequently, secondary factors, myogenin and myogenic regulatory factor 4 (MRF4), regulate terminal differentiation [38]. 

### 5.2. Classification of Skeletal Muscle Fiber Types

By 1873, Ranvier had already categorized muscle into red muscle, with slow contraction, and white muscle, which shows fast contraction. In the 1970s, the classification of type I (slow-twitch red) and type II (fast-twitch white) was proposed based on contractile properties and oxidative capacity. Mitochondria have a critical role in oxidative phosphorylation. The higher density of mitochondria in muscle is correlated with its red color. Some type II fibers possess a faster contractile property than type I, though type I has a higher oxidative capacity. Therefore, type II was subsequently divided into type IIA (fast-twitch red) and type IIB (fast-twitch white). Later, a correlation of myosin heavy chain (MHC) isoform expression and contractile property with myofibrillar ATPase activity was identified. The new type IIx MHC protein was also found. Type IIb MHC is not typically expressed in human skeletal muscle. Currently, fiber types are classified based on MHC isoforms into type I (slow-red), type IIA (fast-red), type IIB (fast-white in rodents), and type IIX (fast-white in humans) (Table 1) [41,42,43,44].

Basically, human skeletal muscles consist of a mixture of muscle fiber types. Some are characterized as type I fiber-dominant (soleus muscle) or type II fiber-dominant (triceps brachii muscle) [43]. The distribution of muscle fiber types in an individual may change with various stimuli such as exercise. For example, muscle-biopsy specimens from elite sprinters were found have an increased number of type II myofibers, in contrast to an increased number of type I myofibers in those from distance runners [2].

Although this classification and nomenclature for skeletal muscle may be insufficient, it is very convenient to use when communicating and sharing findings with other researchers in various fields [42].

## 6. Skeletal Muscle Changes Related to Aging and Damage by CKD

### 6.1. Age-Related Changes of Skeletal Muscle

Aging is a well-known risk factor for loss of skeletal muscle mass. Using magnetic resonance imaging (MRI), skeletal muscle mass and distribution were evaluated in 268 men and 200 women aged 18–88 years [45]. Findings of that quantitative approach showed an apparent decrease in skeletal muscle mass in healthy subjects 60 years of age and older, while notable results indicated a prominent age-related decline in muscle mass in the lower limbs [45]. Aging atrophy seems to be associated with reductions in both number and size of muscle fibers, though that is not well understood [46]. To examine the influence of aging on muscle fiber number and size, quadriceps muscle biopsies, as well as DXA and computed tomography (CT) examinations were performed in healthy young and older males and females [47]. Skeletal muscle mass in the males was significantly greater as compared to that in the females. Furthermore, lean leg mass was significantly lower in the elderly than the young group, while whole-body lean mass did not differ between them. In line with those findings, cross-sectional area (CSA), evaluated based on the quadriceps muscle biopsy results, was smaller in the older as compared to the young group. Especially, type II muscle fiber size was substantially smaller in the older subjects, with a tendency for smaller type I muscle fibers. The calculated number of fibers in the quadriceps did not differ between the groups examined in that study. Therefore, it was concluded that age-related decline in skeletal muscle is mainly due to a reduction in type II muscle fiber size [47]. A contrasting example may be changes in skeletal muscle after a spinal cord injury (SCI), which can cause a fiber-type transformation. In patients treated for SCI, fiber-type shifts from type I or type IIA to type IIX were observed [43]. Interestingly, the transformation pattern in individuals with an SCI is opposite of that induced by aging.

Satellite cells have critical roles for repair and hypertrophy of skeletal muscle, and fiber characteristics and fiber type-specific content related to those cells were examined in young and elderly males [48]. As noted in the report mentioned above [47], biopsy specimens in the vastus lateralis showed that type II, but not type I, muscle fiber CSA was significantly smaller in the older as compared with the young subjects. Satellite cell content determined based on Pax 7-positive cells was also examined, with no difference found between the young and older groups for that content in type I muscle fibers. In contrast, satellite cell content in the type II muscle fibers was significantly lower in the older subjects. These findings suggest that an age-related decline in satellite cell content may be associated with type II muscle fiber atrophy and loss of skeletal muscle mass in older individuals [48].

### 6.2. Histopathological Changes of Skeletal Muscle in Advanced CKD/HD Cases

It seems that skeletal muscle atrophy should be expected in patients with advanced CKD/HD. However, in a study of 13 patients with advanced CKD, type II fiber areas tended to be smaller as compared to the control group, though the difference was not significant [49]. In another study, mean sizes of type I and II fibers in eight patients on dialysis or with kidney transplantation were not different from those in the controls [50], while a different report noted that while type II fibers were slightly but not significantly smaller in 12 dialysis patients as compared to the controls [51]. Rather unexpectedly, no clear findings have yet been presented to support greater muscle atrophy in patients with advanced CKD/HD as compared to control subjects.

A precise muscle biopsy (vastus lateralis)-based study was performed with relatively large groups of subjects (60 patients on HD, 21 controls) [52]. Surprisingly, the mean CSA values for type I, IIA, and IIX fibers were 33%, 26%, and 28%, respectively, greater in the HD patients as compared with the controls [52]. Since those muscle biopsy-specimens were obtained from patients one day after undergoing HD, the greater CSA size might be explained by interstitial edema of skeletal muscle. Another interesting finding in that study showed that the activity of succinate dehydrogenase, a mitochondrial oxidative enzyme, was decreased in muscle fibers of patients undergoing HD. Ultrastructural analysis also revealed swollen mitochondria in the HD patients. In addition, capillary density, evaluated by the number of capillaries per square millimeter of muscle area, was significantly reduced by 34% in the HD patients as compared with the control group [52]. Although interpretation of their results showing impaired oxidative capacity and reduced capillary density in the skeletal muscle of HD patients is difficult, except for enlargement of muscle fibers possibly due to edema, they seem to indicate that impaired energy production together with reduced oxygen supply and substrates may lead to skeletal muscle dysfunction.

Nevertheless, it is also important to note that qualitative changes may veil actual quantity of skeletal muscle. Histopathological findings showing substitution of contractile muscle fibers because of fat infiltration and/or fibrosis may indicate muscle dysfunction, even when muscle mass is similar. Especially, myosteatosis, equivalent to intermuscular adipose tissue, has attracted much attention [17]. Since skeletal muscle attenuation determined using CT is significantly associated with muscle lipid content based on histological findings with oil red O staining and muscle triglyceride measurements in biopsy specimens [53], myosteatosis was previously evaluated indirectly with CT or MRI results. Myofibrosis, pathological fibrosis in skeletal muscle, is thought to accompany myosteatosis and may be the result of various events including injury, inflammation, and degeneration. Recent technological innovations have made it possible to determine myosteatosis and/or myofibrosis using ultrasound. With this method, it has been reported that myosteatosis and/or myofibrosis were negatively correlated with physiological performance in CKD patients [18]. Furthermore, since those might be involved in systemic metabolic and inflammatory disorders, as well as subsequent mortality, additional investigations are required to confirm the impact of myosteatosis and myofibrosis on sarcopenia in advanced CKD/HD.

## 7. Aging Kidneys in CKD Patients

Various organs including the kidneys are involved in its onset and development of sarcopenia [3,10,19,28]. When lifespans were short, kidney failure might not have been a major issue, as most humans experienced mortality first. However, the recent super-aging society is faced with new disorders such as CKD.

The prevalence of CKD throughout the world including Japan increases with age. It has been reported that 13% of the Japanese general population (13.3 million people) has CKD when that is defined as glomerular filtration rate (GFR) < 60 mL/min/1.73 m^2^) [54]. Along with aging, the prevalence of CKD gradually increases, with four out of ten individuals aged 80 and older affected. Although it seems to be a natural occurrence, age-related structural changes in kidneys have not been clarified. Recent studies have reported macroanatomical changes based on results obtained with imaging modalities such as CT [55] and microanatomical changes based on kidney biopsy findings [56,57] in living kidney donors who had no obvious kidney disease.

Contrast-enhanced CT imaging can be used to measure not only total kidney volume, but also cortical and medullary volumes separately. Cortical volume seems to decrease with age, whereas medullary volume increases until age 60, which may be compensatory, and then remains unchanged thereafter [55]. Thus, total kidney volume does not change until age 60 and then subsequently decreases [55]. Since the cortex includes nephron and proximal tubules, it would be interesting to know whether age-related loss of cortical volume reflects nephron loss and/or decrease in GFR, though those have yet to be clarified [58].

Using renal biopsy samples obtained from living kidney donors, anatomical changes were examined [56,57]. Nephrosclerosis, a major age-related change, includes glomerulosclerosis, tubular atrophy, interstitial fibrosis, and arteriosclerosis. Sclerosis score was determined as the total number of abnormalities, including any type of global glomerulosclerosis, tubular atrophy, and arteriosclerosis, and interstitial fibrosis > 5%. When nephrosclerosis was defined as two or more of the total four, an increasing prevalence of nephrosclerosis associated with aging was observed in 2.7% of the subjects younger than 29 years and in 73% of those aged greater than 70 [57]. Decreased glomerular density, calculated based on the number of glomeruli divided by the area of the cortex, was also associated with older age in living donors [56]. These findings suggest that conditions related to aging kidneys and consequent CKD-related sarcopenia are inevitable in super-aging societies.

## 8. Factors Possibly Affecting Muscle Wasting in Advanced CKD/HD

Large numbers of different factors and/or mechanisms including persistent inflammation are considered to be involved with age-related muscle derangement (sarcopenia) in CKD/HD patients, with both expected and unexpected findings reported [4,16,59,60] [61], and new information presented even in the last few years. For example, uremic toxins such as indoxyl sulfate and p-cresol were shown to impair myogenic differentiation of cultured C2C12 skeletal muscle cells [62]. Inorganic phosphate (Pi) was also found to decrease myogenic differentiation in vitro and promoted muscle atrophy in CKD mice [63]. In a randomized controlled study of pre-dialysis CKD patients, oral sodium bicarbonate achieved a serum level of ~24 mEq/L for preserved muscle mass [64]. These recent reports have provided new insights in this field. Nevertheless, systematic and comprehensive classifications of cause and etiology are still required, though currently very difficult to establish. In the following, clinically relevant and topical factors that have been relatively well investigated and established to some extent in this field will be discussed (Figure 3).

### 8.1. Vitamin D

Vitamin D is one of the critical components in CKD-mineral and bone disorder (CKD-MBD) [65]. It regulates not only calcium homeostasis and bone metabolism, but also skeletal muscle metabolism [66,67]. Furthermore, the active form of 1α,25-dihydroxyvitamin D [1,25(OH)_2_D] binds to vitamin D receptor (VDR), and can exert diverse biological effects through genomic and non-genomic activities, while expression of VDR in both animal and human muscle tissues was identified [66,67]. Another investigation showed that though synthesis of 1,25(OH)_2_D from 25(OH)D is mediated by mitochondrial 1α-hydroxylase encoded by the *Cyp27b1* gene, predominantly expressed in the kidneys, C2C12 myoblasts and myotubes expressed both VDR and CYP27B1 [68]. Therefore, skeletal muscle cells seem to possess machinery for response to vitamin D. In fact, addition of 1,25(OH)_2_D to C2C12 myoblasts increased VDR expression, decreased cell proliferation, and promoted myogenic differentiation [69]. Those authors also showed that 1,25(OH)_2_D increased expression of MyoD and subsequently suppressed myostatin in a time-dependent manner, and finally increased the diameter and size of MHC type II-positive cells. Findings obtained in mice with deletion of the vitamin D receptor (VDRKO) also support the role of vitamin D in skeletal muscle maintenance. The VDRKO group showed smaller muscle mass and weaker grip strength as compared with the controls [70]. Another study reported that mice following VDRKO had smaller diameter muscle fibers with an aberrant reversed higher expression of myogenic differentiation factors as compared to wild-type mice, suggesting a physiological role of vitamin D through temporal up-regulation of myogenic transcription factors [71]. In this context, vitamin D (calcitriol) also seems to antagonize CKD-induced skeletal muscle changes [72]. That study reported that high-phosphate diet accelerated skeletal muscle changes in CKD rats (5/6 nephrectomy) as compared with a standard diet. Furthermore, low dose calcitriol attenuated adverse skeletal muscle changes in CKD rats that received a high phosphate diet. Interestingly, calcitriol improved the number of capillaries in contact with muscle fibers. Although the precise mechanisms are still unclear, vitamin D may have pleiotropic effects on skeletal muscle maintenance in patients with CKD.

A link between vitamin D and skeletal muscle is probable in rodents, while related effects on and mechanisms in human skeletal muscle remain to be established. It has been reported that a vitamin D system was detected in human muscle precursor cells, though was low in adult skeletal muscle [73]. They also noted that vitamin D seems to promote myoblast self-renewal and maintain the satellite stem cell pool through modulation of the forkhead box O (FOXO) 3 and Notch signaling pathways. In contrast to rodents, how vitamin D can affect skeletal muscle maintenance in humans remains to be elucidated.

Vitamin D deficiency in humans is evaluated based on the storage form of vitamin D, 25(OH)D, and a strong association of vitamin D deficiency with muscle dysfunction has been shown [74]. Generally, optimal musculoskeletal benefits occur at 25(OH)D levels above 30 ng/mL. In contrast, in evaluated biopsy specimens, vitamin D deficiency was found to be correlated with skeletal muscle dysfunction and predominantly associated with type II muscle fiber atrophy [66]. More directly, an examination of the correlation between vitamin D deficiency and sarcopenia, defined based on appendicular skeletal muscle mass divided by body weight less than two standard deviation (SD) below the sex-specific mean for young adults, performed in 3169 Korean participants showed that the mean 25(OH)D concentration was significantly lower in those with than without sarcopenia [75].

### 8.2. Myostatin

Myostatin, also known as growth development factor-8 (GDF-8), has been identified as a negative regulator of skeletal muscle growth [76,77]. This newly established factor is mainly secreted by muscle cells and belongs to the transforming growth factor-β (TGF-β) superfamily. Myostatin binds to its cognate receptor, activin type II B receptor (ActRIIB), and exerts diverse effects. Upon activation of Smad2/Smad3 and dephosphorylation of Akt, muscle protein ubiquitination and degradation by proteasomes and autophagy are induced through Atrogin-1 and muscle ring finger 1 (MuRF1), leading to increased protein degradation [60,78]. Myostatin can also inhibit mammalian target of rapamycin complex 1 (mTORC1), one of the key molecules in protein synthesis, resulting in a decrease in that process [60,78]. Additionally, myostatin-induced apoptosis was shown to occur via activation of the p38-caspase pathway [78].

Although myostatin might be involved in development of sarcopenia in CKD/HD, limited data are available. Cy/+ rats, which develop advanced CKD due to a genetic defect, were reported to show a progressive decline in muscle function [79]. Additionally, as compared to control rats, Cy/+ rats had a significantly higher serum level of myostatin and increased expression of myostatin in skeletal muscle, along with higher indices of oxidative stress [80]. In half-nephrectomized mice, indoxyl sulfate, a uremic toxin, induced skeletal muscle weight loss, which was accompanied by expression of myostatin and atrogin-1 in addition to increased production of inflammatory cytokines in skeletal muscle [81].

In humans, it has been speculated that the plasma or serum concentration of myostatin can be used as a biomarker of muscle wasting. However, inconsistent results have been presented i.e., both positive and negative, or no significant correlation between myostatin and muscle mass and/or muscle strength [82]. Nevertheless, that report noted that emerging evidence suggests that myostatin is influenced by various factors such as age, gender, and physical activity as well as a wide range of disorders including heart failure, metabolic syndrome, CKD, and inflammatory diseases. Another reason for conflicting results may be the different assay techniques utilized. Generally, the level of myostatin in CKD/HD patients seems to be higher than that in healthy controls [78]. A recent study showed that myostatin was positively associated with muscle strength, evaluated by handgrip strength, as well as muscle mass in patients undergoing HD [83]. Moreover, a lower level of myostatin was demonstrated to be a significant predictor of one-year mortality in that study. Additional investigations are needed to confirm whether myostatin is a biomarker for sarcopenia in advanced cases of CKD/HD.

### 8.3. IGF-I

Insulin-like growth factor-I (IGF-I), well known to be associated with muscle mass and fiber size [84,85], binds to its receptor and exerts biological effects. Although IGF-I shares common intracellular signaling pathway with insulin, it appears to have more growth effects than metabolic effects as compared to insulin, though the mechanisms are unknown [86]. In protein synthesis, IGF-I-induced activation of mTORC1 through phosphatidylinositol-3-kinase (PI3K)/Akt is necessary. At the same time, activation of the PI3K/Akt pathway results in phosphorylation of FOXO proteins. It has also been reported that IGF-I suppressed ubiquitin ligases, such as Atrogin-1 and MuRF1, via Akt-mediated inhibition of FOXO1, suggesting antagonizing effects of IGF-I in skeletal muscle catabolism [87,88].

To examine growth factors including IGF-I, vastus lateralis muscle biopsies were performed in 55 patients undergoing HD and 21 healthy subjects. As expected, mRNA for IGF-I/IGF-I receptor was decreased in skeletal muscle from the HD patients as compared with the healthy controls. However, protein levels for muscle IGF-I and serum IGF-II were increased [89]. Although IGF-I plays a critical role in muscle maintenance, its action mechanisms may be complicated, i.e., transient binding of the IGF-I receptor, degradation of the IGF-I/IGF-I receptor, local concentrations of IGF-I, and the influence of IGF-I binding proteins.

In addition to direct effects on skeletal muscle, IGF-I seems to be involved in CKD-induced dysfunction of satellite cells. It has been reported that isolated Pax-7 positive cells (satellite cells) from CKD mice obtained by use of a subtotal nephrectomy had lower levels of MyoD expression and showed suppressed myotube formation [90]. Additionally, CKD mice showed delayed regeneration of injured muscle, and decreased MyoD and myogenin expression. IGF-I increased the expression of myogenic genes in response to injury in satellite cells from the control group but not in those from CKD mice. Therefore, impaired IGF-I signaling may be associated with satellite cell dysfunction as well as protein catabolism in CKD-related sarcopenia.

### 8.4. Insulin and Glucose (Hyperglycemia)

Insulin is one of most important hormones in human body and its metabolic disarrangement in skeletal muscle is profoundly associated with the etiology of diabetes [91,92,93,94]. In addition to metabolic effects, insulin has also been shown to be involved in protein anabolism in vitro [95], as well as in vivo in rats [96,97], and humans [98]. However, the direct effects of insulin on protein metabolism in human skeletal muscle is poorly understood as compared with glucose and lipid metabolism.

A detailed study provided new insights in regard to insulin-mediated protein synthesis in human skeletal muscle [99]. Insulin was infused within a physiological range in 19 young subjects, then protein synthesis was evaluated in muscle biopsy specimens to determine uptake of isotope-labeled phenylalanine, which was not intracellularly oxidized. At the same time, blood flow was measured using indocyanine green. Insulin can stimulate muscle protein synthesis. Interestingly, increases in blood flow and amino acid delivery to skeletal muscle were found to be critical factors in insulin-induced protein synthesis in that study. Using a similar technique, insulin-stimulated protein synthesis in skeletal muscle was compared between young and older healthy subjects [100]. In the older group, protein synthesis was found to be resistant to insulin, suggesting that aging is a critical risk factor for maintenance of skeletal muscle via age-related insulin resistance.

In addition to insulin, glucose may be involved in skeletal muscle maintenance. A recent study demonstrated suppressed proliferation of satellite cells in high-glucose culture media [101]. In contrast, glucose restriction led to a relative increase in Pax7-positive/MyoD-negative cells, which were equivalent to reserve cells for self-renewal. Those findings suggest that hyperglycemia might inhibit regeneration of skeletal muscles and accelerate sarcopenia in patients with diabetes. To examine whether hyperglycemia is associated with sarcopenia, a multicenter cross-sectional study was performed in 746 patients with type 2 diabetes (T2D) and 2067 other older participants [102]. Hyperglycemia represented as HbA1c was found to be an independent contributor to the presence of sarcopenia, especially in the non-obese subjects. Interestingly, HbA1c level was specifically associated with low skeletal mass index (SMI) rather than weak grip strength or slow gait speed. Therefore, appropriate glycemic control should be considered as a requirement for prevention of sarcopenia, probably in HD patients with diabetes.

### 8.5. Physical Inactivity and Sedentary Lifestyle in Older Patients with CKD and HD

Physical activity gradually decreases with aging and incidental HD patients show a greater tendency for physical inactivity. The association of physical inactivity with malnutritional status was examined in HD patients and healthy sedentary controls [103]. Both physical activity, measured by a three-dimensional accelerometer, and energy expenditure, evaluated by questionnaire, were lower in the patients and the difference between those groups increased with advancing age. Additionally, physical activity in the HD patients was associated with serum albumin and creatinine levels, as well as phase angle derived from BIA. These findings suggest an association between physical inactivity and malnutritional status including muscle wasting. The benefit of regular physical activity on mortality was examined in a national cohort of new patients with end-stage kidney disease in the United States, which showed that mortality risk was lower in those who exercised 2–3 or 4–5 times a week, suggesting an association of physical activity with survival in dialysis patients [104]. In addition to aging, the presence of advanced CKD/HD may have a strong impact on the skeletal muscle system, leading to significant exercise intolerance [105]. To break the vicious cycle between physical inactivity and mortality, exercise may be one of most promising and hopeful approaches, which will be discussed later.

### 8.6. Mitochondria

Mitochondria, small membrane-bound organelles, generate a significant amount of energy in the form of ATP. Due to its heavy demand for energy as a motor organ, skeletal muscle has a large number of mitochondria. Mitochondrial dynamics are regulated by fine balance between fusion and fission [106,107]. For mitochondrial fusion, the mitofusins Mfn1 and Mfn2, mitochondrial GTPases, are essential [108], while on the other hand, fission protein 1 (Fis1) and dynamin-related protein 1 (DRP-1) maintain mitochondrial homeostasis through appropriate fission (fragmentation) [106]. Mitochondrial quality control is necessary for structural and functional integrity [109]. A critical contributing factor to mitochondrial decay, closely linked to sarcopenia, is aging, which has been shown to be related to decreases in mitochondrial content, enzyme activities such as cytochrome c oxidase and citrate synthase, and oxidative capacity in human skeletal muscle [110]. Impaired mitochondrial respiration leads to reactive oxygen species (ROS) production and increased mitochondrial DNA mutations. In association with that, defective mitochondrial quality control has also been detected in aged muscle and hyperfusion is known to be associated with the appearance of enlarged mitochondria, which cannot be effectively eliminated and results in increased ROS production [109]. In contrast, hyperfission is also observed during the progression of sarcopenia. ROS production by derangements in fusion-fission results in muscle protein breakdown through ubiquitin ligases such as atrogin-1 and Murf-1 [109].

In addition to aging, CKD appears to be involved in mitochondrial dysfunction. Microarray analysis of CKD patients including those undergoing HD suggested an impaired mitochondrial respiratory system and related oxidative stress [111]. More directly, mitochondrial function was evaluated using ^31^P magnetic resonance spectroscopy to determine phosphocreatine recovery time constant in patients with CKD and those on HD [112]. In that study, faster phosphocreatine recovery kinetics (shorter time constant) indicated better mitochondrial function, and a prolonged phosphocreatine recovery time was found in HD and CKD patients as compared with controls. Mitochondrial dysfunction is correlated with poor physical performance, increased intermuscular adipose tissue, and increased markers of inflammation and oxidative stress. Interestingly, that study also found that DRP-1, a marker of mitochondrial fission, was up-regulated in skeletal muscle of HD patients as compared to controls. Therefore, correction of mitochondrial dysfunction is an attractive therapeutic approach for elderly patients with advanced CKD/HD.

## 9. Management of Skeletal Muscle Maintenance in HD Patients

### 9.1. Vitamin D

Based on findings described above showing a positive correlation of vitamin D deficiency with skeletal muscle dysfunction, supplementation with vitamin D is an attractive interventional approach against sarcopenia. In fact, two meta-analysis reports have suggested the beneficial effects of vitamin D supplementation on muscle strength and function in older individuals with a low serum 25(OH)D level at the baseline (vitamin D deficiency) [113,114]. In the same context, a double-blind randomized placebo-controlled trial was performed to explore the effects of 12 months of vitamin D supplementation on lower-extremity power (primary endpoint) and function in healthy community-dwelling elderly subjects [115]. Unexpectedly, there were no differences for lower-extremity power, strength, or lean mass found between the placebo and vitamin D groups. However, the period of intervention, subject background details, and target level of vitamin D should be considered when interpreting those results. Notably, HD patients tend to show severe abnormal vitamin D metabolism as compared to healthy elderly individuals.

A retrospective cross-sectional study was performed to examine the association of vitamin D treatment with muscle mass and function in HD patients, in which muscle size was evaluated by MRI and strength in the lower limbs was measured, with or without treatment with active vitamin D (calcitriol or paricalcitol) [116]. They found that vitamin D treatment was associated with greater muscle size and strength. Additionally, two double-blind randomized placebo-controlled trial were performed to investigate the pleiotropic effects of vitamin D (cholecalciferol) in HD patients [117,118]. In both, the results of various muscle function tests including muscle strength were examined as clinical endpoints. No significant effect of vitamin D supplementation was found in those tests. Although the studies were well-designed, the sample sizes were small (*n* = 52, 60, respectively) and the intervention periods short (eight weeks and six months, respectively), which are limitations. Under these conditions, it might be difficult to confirm the effects of vitamin D supplementation on muscle function. In addition, muscle mass was not evaluated. In future, randomized trials with larger sample sizes and longer periods that focus on sarcopenia as the primary endpoint will be needed.

### 9.2. Myostatin

Myostatin is an attractive therapeutic target for treatment of age- and CKD-related sarcopenia [16,60]. Two strategies to inhibit myostatin pathways are under development, one is a blockade caused by direct binding to myostatin itself and the other is inhibition of the myostatin-ActRIIB complex [78]. One of the candidates is the monoclonal antibody LY2495655, which binds and neutralizes myostatin, and was examined in a phase II study [119]. This antibody was found to have a relationship with lean mass and partially improved functional measures of muscle power in older frail individuals. On the other hand, results of another phase II trial showed no significant effect of LY2495655 on lean body mass in patients undergoing elective total hip arthroplasty [120]. Similarly, the efficacy of bimagrumab, an anti-ActRIIB antibody, was examined in older adults with sarcopenia in a phase II trial [121], and found to increase muscle mass and strength.

Administration of an anti-myostatin peptibody in CKD mice reversed loss of muscle mass and suppressed circulating inflammatory cytokines [122]. Additionally, it has recently been reported that formononetin, a bioactive isoflavone compound, ameliorates muscle atrophy in CKD rats by antagonizing myostatin [123]. If myostatin-mediated treatment is found to be effective in older patients with advanced CKD/HD, important points for consideration will be whether it improves not only muscle mass but also muscle strength.

### 9.3. Insulin and Anti-Diabetic Treatments

Since insulin deficiency is an independent risk factor for sarcopenia, the next question may be whether insulin treatment can protect against sarcopenia. A retrospective observational study examined the association of insulin treatment with SMI, calculated as appendicular muscle mass using DXA divided by the square of height, in 312 patients with T2D [124]. Insulin treatment was shown to be protective against annual decline in SMI after adjusting for various factors. In a propensity score-matched cohort in the same study, the annual change in SMI was greater in the insulin-treated than non-insulin-treated group. These findings suggest that insulin has a critical role not only in glucose metabolism but also maintenance of skeletal muscle mass in patients with diabetes.

Few studies have investigated the association of oral anti-diabetic drugs with sarcopenia. Considering the possible efficacy of insulin, dipeptidyl peptidase 4 inhibitors (DPP4-I) are good candidates for sarcopenia treatment because they stimulate insulin secretion in a blood glucose-level dependent manner. DPP4-I is preferrable in regard to its efficacy, low risk of hypoglycemia, and good tolerability in elderly patients with T2D. In line with those factors, the association of sarcopenia, diagnosed according to the EWGSOP criteria, with DPP4-I was examined in 80 elderly patients with T2D [125]. The participants were divided into the sulfonylurea (*n* = 43) and DPP4-I (*n* = 37) groups, and followed for at least 24 months. The DPP4-I group showed greater muscle mass, as well as better muscle strength and physical performance as compared with sulfonylurea group. Since physical activity and nutritional status of those participants were relatively stable, the authors suggested that the better sarcopenic parameters noted were mainly due to treatment effects. Similarly, the association of use of DPP4-I with loss of muscle mass was examined in a retrospective observational study that included 105 patients with T2D [126]. Propensity-score matching analysis was performed to remove bias. SMI, determined using DXA, was evaluated and its annual change was significantly higher in patients with as compared to those without DPP4-I [126]. On the other hand, there were no significant differences in regard to changes in visceral and subcutaneous fat area between those groups.

Most oral anti-diabetic drugs are contraindicated or restricted in HD patients, as renal dysfunction brings about profound pharmacokinetic abnormalities [127]. Among those drugs, DPP4-I are available for HD patients [128,129]. Although no report of the efficacy of DPP4-I on sarcopenia in HD patients has been presented, those inhibitors may be considered as candidates for preventing age-related loss of muscle mass in patients with T2D.

### 9.4. Nutrition

Adequate intake of nutrients is necessary to prevent the onset and progression of sarcopenia, PEW, and frailty. In contrast to common agreement regarding adequate energy intake (generally, 30–35 kcal/kg/day), how much dietary protein intake is ideal for CKD and dialysis patients remains controversial. Typically, protein intake of 0.6–0.8 g/kg/day is recommended for advanced pre-dialysis patients from the viewpoint of uremia and kidney protection. On the other hand, that recommendation is usually ⁓1.2 g/kg/day in dialysis patients to preserve muscle mass [6]. When a standard prevention approach is not enough, nutritional supplementation is considered, with many methods available, including oral supplementation, intradialytic parental nutrition, and tube feeding [4,6,61,130].

When focusing on maintenance of skeletal muscle in CKD patients, especially those undergoing HD, amino acid supplementation is reasonable. Amino acids are not only precursors for muscle protein synthesis but also stimulators of intracellular signaling. Notably, among essential amino acids, leucin has a distinct anabolic action via the mTORC1 pathway [131]. As an intracellular regulator, sestrin2 has received much attention and it has been reported that its leucine-binding capacity is necessary for leucin-induced activation of mTORC1, indicating that sestrin2 functions as a sensor of leucin and inhibitor of mTORC1 [132]. A recent systematic review reported no clear effects of supplementation with essential amino acids on muscle mass, muscle strength, or physical performance, whereas a significant effect of leucine on muscle mass was shown in subjects with sarcopenia as compared with healthy subjects [133].

Another possible supplementation strategy might be use of β-hydroxy-β-methylbutyrate (HMB), a metabolite of leucin [134]. To examine its effects on body composition, bone density, strength, physical function, and other such parameters in HD patients, a double-blind placebo-controlled randomized trial was performed over a period of six months, though no significant effects of HMB supplementation were observed [135]. Based on findings presented to date, leucin may be one of the good candidates for nutritional supplementation to maintain skeletal muscle. As described, following exercise or physical activity, nutritional supplementation should be able to provide synergistic beneficial effects on maintenance of skeletal muscle in HD patients [136].

### 9.5. Exercise

Skeletal muscle is characterized as a dynamic organ, and muscle contractions may have a special role in homeostasis of skeletal muscle separate from hormones and cytokines [137,138]. Thus, to overcome aging-related sarcopenia, it may be a powerful approach. Especially, aerobic exercise has been shown to improve insulin-stimulated muscle protein synthesis in elderly individuals, though aging dampens the anabolic effects of insulin on skeletal muscle [139]. In patients undergoing HD, aerobic exercise seemed to increase CSA, while an interesting finding also noted was an improvement in exercise-induced capillarization in skeletal muscle [140]. In combination with previous findings [99,139], it is considered that an increase in blood flow in skeletal muscle in response to aerobic exercise and insulin may contribute to maintenance of muscle mass and function.

Exercise may also counteract CKD-induced muscle wasting. Using CKD mice induced by a subtotal nephrectomy, the effects of resistance exercise (muscle overload) and endurance training (treadmill running) on skeletal muscle wasting were examined [141]. Although both types of exercises suppressed CKD-induced muscle protein degradation, the specific effects differed between them, as improved muscle protein synthesis was observed in mice with resistance exercise but not in those with endurance training. Additionally, those results showed that resistance exercise counteracted CKD-induced suppression of phosphorylation of S6K and mTORC1 in muscle, resulting in maintenance of protein synthesis. In these mice, resistance exercise but not endurance training increased the number of muscle progenitor cells. Together, these findings suggest that different types of exercise can provide various effects related to skeletal muscle maintenance.

Muscle biopsy procedures were performed in dialysis patients (18 HD, three peritoneal dialysis) to examine the effects of resistance training on skeletal muscle [142]. The number of satellite cells in type I muscle fibers was increased after 16 weeks of resistance training performed three times a week, whereas those in type II fibers remained unchanged. However, an increase in the myonuclear contents of the type II fibers was observed. Satellite cells may directly differentiate into myonuclei without asymmetric division, though that does not seem likely. Although interpretation of their observations is difficult, one of the most important findings might be increased muscle strength, which can be determined by torque measurements in clinical settings.

In addition to exercise, that in combination with dietary protein supplementation might exert synergistic effects on maintenance of skeletal muscle mass and function in various individuals [143,144]. As described above, leucin may have a key role. To determine the effects of leucin ingestion on muscle protein synthesis after resistance exercise in elderly male subjects, the results of infusion of isotope-labeled phenylalanine with leucin shown in muscle biopsy specimens were examined [145]. At 24 h after exercise, protein synthesis remained elevated in the leucin group as compared with the controls, which indicated a potential impact of leucin in combination with exercise for muscle maintenance in the older individuals. Additionally, it is possible that a combination of exercise with nutritional support, especially leucin, has beneficial effects in HD patients [136], though clear evidence remains lacking.

## 10. Conclusions

The modernization of society along with progress in medical treatments have led to new disorders, such as frailty, sarcopenia, and PEW, especially in elderly HD patients. The increased risk of those conditions has led to speculation and reservations regarding patient care. On the other hand, intensive investigations have provided abundant new insights in this field. A broad range of communication, from bench to bedside, and the reverse as well, will be required. In clinical practice, incorporation of various therapeutic options should also be considered, as such fusion will result in a much better outcome including extended healthy life expectancy in elderly patients receiving HD treatments.

## Figures and Tables

**Figure 1 nutrients-13-01538-f001:**
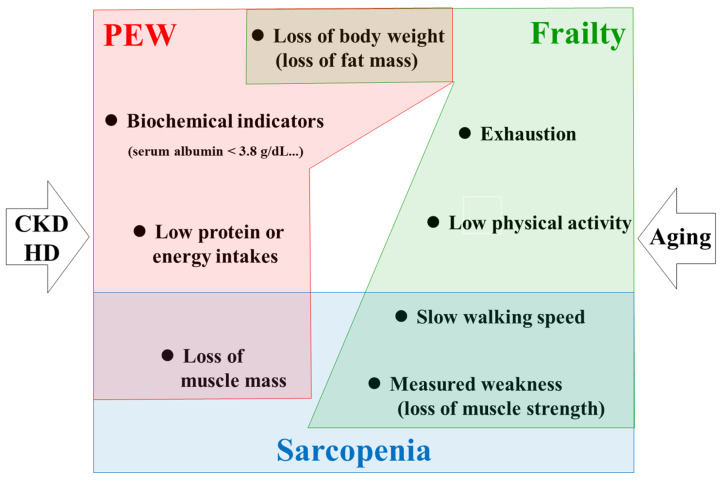
Conceptual overlapping among frailty, sarcopenia, and PEW. Aging accelerates frailty and sarcopenia. Advanced CKD/HD is profoundly associated with PEW. Elderly HD patients have the highest risk for these pathological conditions. Among these conditions, skeletal muscle derangement is a common component. CKD, chronic kidney disease; HD, hemodialysis; PEW, protein-energy wasting.

**Figure 2 nutrients-13-01538-f002:**
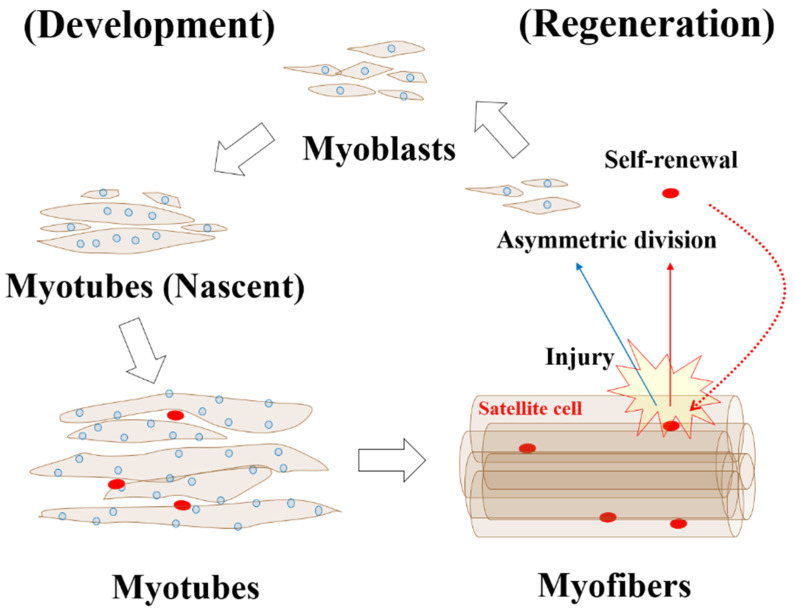
Muscle development and regeneration. During the early stage of development, myoblasts fuse to form multinucleated nascent myotubes that show central nucleation. In a later stage, nuclei migrate to the periphery of mature myotubes, resulting in formation of myofibers. Satellite cells are located between the basal lamina and plasma membrane. For muscle regeneration, satellite cells have a key role. Diverse stimuli including injury activate those cells, which is followed by asymmetric division to conserve the satellite cell pool (self-renewal) and generation of committed myogenic precursors (myoblasts) for regeneration via a process equivalent to development.

**Figure 3 nutrients-13-01538-f003:**
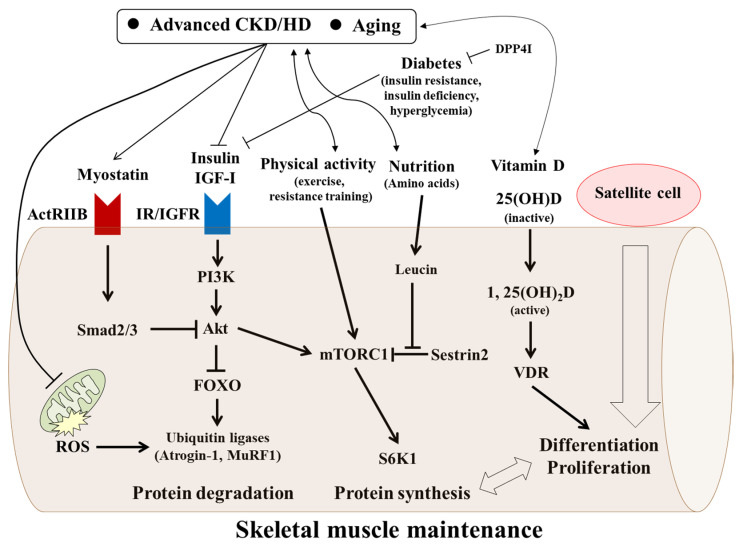
Various factors affecting skeletal muscle maintenance in aged patients with advanced CKD/HD. In aged patients with advanced CKD/HD, various factors have interaction with and transduce their effects intracellularly, thus affecting skeletal muscle maintenance. Insulin and IGF-I positively regulate skeletal muscle maintenance via binding of their cognate receptors. Subsequent activation of mTORC1 through PI3K/Akt is necessary for protein synthesis. Simultaneously, insulin-stimulated PI3K/Akt increases FOXO phosphorylation and subsequent inhibition of its translocation into the nucleus. On the other hand, myostatin, a negative regulator, binds to ActRIIB. Subsequent phosphorylation of Smad2/3 reduces Akt activation and decreases FOXO phosphorylation. Intranuclear translocated FOXO activates transcription of MuRF1 and Atrogin-1, which accelerates protein degradation via the ubiquitin-proteasome pathway. Defective mitochondrial quality control (derangements in fusion-fission) leads to ROS production and results in muscle protein degradation. Vitamin D is involved in muscle differentiation and proliferation by binding VDR, which is accompanied by interaction with muscle protein metabolism. Among amino acids, leucin activates mTORC1 by binding to Sestrin2, leading to protein synthesis. In addition, satellite cells, which can be modulated by the various aforementioned factors, may contribute to muscle maintenance. CKD, chronic kidney disease; HD, hemodialysis; ActRIIB, activin type II B receptor; IGF-I, insulin-like growth factor I; mTORC1, mammalian target of rapamycin complex 1; PI3K, phosphatidylinositol-3-kinase; FOXO, forkhead box O; MuRF1, muscle ring finger 1; ROS, reactive oxygen species; VDR, vitamin D receptor; DPP4-I; dipeptidyl peptidase 4 inhibitors.

**Table 1 nutrients-13-01538-t001:** Classification of skeletal muscle fiber types.

	Type I	Type IIA	Type IIX (Human)	Type IIB (Mouse, Rat)
Anatomical color	Red	Red	White	White
Contractile speed	Slow-twitch	Fast-twitch	Fast-twitch	Fast-twitch
Myosin heavy chain isoform	Type I	Type IIa	Type IIx	Type IIb
Metabolic	Oxidative	Oxidative	Glycolytic	Glycolytic
Myofibrillar ATPase activity	Low	High	High	High
Mitochondrial density	High	High	Medium	Low
Fatigue	Resistant	Resistant	Fast	Fast

Presently, skeletal muscle fiber types are classified using histochemical methods and based on metabolic differences. Most skeletal muscle tissues consist of heterogenous fiber types that show a mosaic pattern. In response to various stimuli and circumstances, muscle fiber types are transformed.

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
