# Peer review of "Maintenance of Skeletal Muscle to Counteract Sarcopenia in Patients with Advanced Chronic Kidney Disease and Especially Those Undergoing Hemodialysis"

_nutrients, 2021, doi:10.3390/nu13051538_

Round 1

Reviewer 1 Report

Excellent work about the interrelation of sarcopenia, frailty and PEW in patients with CKD and HD. The methodology is correct and offer a good physiopathology and cellular mechanism and the assessment of muscle mass (detection methods, differences according to races) and modifications with the new consensus of the European group of 2019.

Good squemes and mechanism of myocitoqines production.

It is also worth mentioning the complete approach to the alternatives of expendable life habits with physical activity included.

Reviewer 2 Report

The work deals with the important problem of sarcopenia developing in patients undergoing dialysis. It is a review, written in an understandable way, and the figures it contains make it easier to perceive the content. Nevertheless, while reading it, I noticed some minor shortcomings, which I present below. Before accepting the work for publication, the authors should refer to them and implement a minor correction.

  • The manner of quoting the works in the manuscript should be standardized.Sometimes authors cite works in the following way: [1] [2] [3], and once [1,2].
  • Figures should include explanations of all abbreviations in their captions - this is missing under figure 1.
  • Although the concept of the frailty syndrome has been used in geriatrics for a long time, I would be far from saying that this concept is "easy to understand" (penultimate paragraph on page 2);
  • - MAC is used more often than MAMC
  • Page 7- second paragraph in section 6.2- 4th sentence - unnecessary word "patient" at the end.
  • - Page 10 chapter 8.2- it is worth referring to the work [Baczek J, Silkiewicz M, Wojszel ZB. Myostatin as a Biomarker of Muscle Wasting and other Pathologies-State of the Art and Knowledge Gaps. Nutrients. 2020 Aug 11;12(8):2401. doi: 10.3390/nu12082401. ]

Reviewer 3 Report

This paper is a review on sarcopenia in hemodialysis patients. The paper is very interesting and scientifically valuable. However, the topic is very wide and the paper is sometimes off topic.

For example, the paragraph on kidney aging is very interesting, but has it a place in this paper on muscle abnormalities? The same paragraph on muscle aging could be more appropriate in this paper...

 Another example: in the VitaminD paragraph 8.1, no information is given to make the connection between VitaminD, CKD and sarcopenia, but literature exists on this topic (at least showing the association between muscle mass/strength and VitD concentrations in CKD and HD patients).

The paragraph of the impact of diabetes on muscle (8.5) is not fully in the topic, although I understand that diabtes prevalence is high in CKD.

On top of that, some aspects of muscle abnormalities are missing, specifically a paragraph on mitochondria dysfunction in CKD should be written (see Skeletal Muscle Mitochondrial Dysfunction Is Present in Patients with CKD before Initiation of Maintenance Hemodialysis. Jorge L Gamboa, Baback Roshanravan, Theodore Towse, Chad A Keller, Aaron M Falck, Chang Yu, Walter R Frontera, Nancy J Brown, T Alp Ikizler. Clin J Am Soc Nephrol. 2020 Jul 1;15(7):926-936.).

Major comments

  • In the introduction, page 1, the definition of sarcopenia has been revised in 2019 and the reference 3 cited at the end of first paragraph should be replaced by reference 10: Sarcopenia: revised European consensus on definition and diagnosis. Cruz-Jentoft AJ, Bahat G, Bauer J, Boirie Y, Bruyère O, Cederholm T, Cooper C, Landi F, Rolland Y, Sayer AA, Schneider SM, Sieber CC, Topinkova E, Vandewoude M, Visser M, Zamboni M; Writing Group for the European Working Group on Sarcopenia in Older People 2 (EWGSOP2), and the Extended Group for EWGSOP2. Age Ageing. 2019 Jan 1;48(1):16-31.
  • Paragraphs 3 and 4: The authors report the prevalence of low muscle mass in HD patients, but not the prevalence of low muscle strength. This is important, as a low muscle strength is more frequent than a low muscle mass (see Clin Nutr Stanislas Bataille, Marianne Serveaux, Elisa Carreno, Nathalie Pedinielli, Patrice Darmon, Alain Robert. The diagnosis of sarcopenia is mainly driven by muscle mass in hemodialysis patients. 2017 Dec;36(6):1654-1660.).
  • The difficulties of muscle mass measurement are also due to muscle histological modifications including myofibrosis and myosteatosis: the measurements not only measure the muscle fibers, but alson non-contractile tissue... This point should be highlighted in the paragraph 3 on muscle mass measurement and in paragraph 6.2 on histological abnormalities of muscle during CKD (see: Quality over quantity? Association of skeletal muscle myosteatosis and myofibrosis on physical function in chronic kidney disease. Thomas J Wilkinson, Douglas W Gould, Daniel G D Nixon, Emma L Watson, Alice C Smith. Nephrol Dial Transplant. 2019 Aug 1;34(8):1344-1353.)
  • Paragraph 8.2: Not only plasma concentration of myostatin in CKD/HD semmes to be higher thant in healthy controls, but it is also correlated to muscle strength and mass. This should be stated (see Delanaye P, Bataille S, Quinonez K, Buckinx F, Warling X, Krzesinski JM, Pottel H, Burtey S, Bruyère O, Cavalier E. Myostatin and Insulin-Like Growth Factor 1 Are Biomarkers of Muscle Strength, Muscle Mass, and Mortality in Patients on Hemodialysis. J Ren Nutr. 2019 Nov;29(6):511-520.).
  • In the paragraph on vitamin D supplementation, two randomized studies performed in HD patients should be described and cited: Marckmann, P. et al. Randomized controlled trial of cholecalciferol supplementation in chronic kidney disease patients with hypovitaminosis D. Nephrol. Dial. Transplant. Off. Publ. Eur. Dial. Transpl. Assoc. - Eur. Ren. Assoc. 27, 3523–3531 (2012). And Hewitt, N. A., O’Connor, A. A., O’Shaughnessy, D. V. & Elder, G. J. Effects of cholecalciferol on functional, biochemical, vascular, and quality of life outcomes in hemodialysis patients. Clin. J. Am. Soc. Nephrol. CJASN 8, 1143–1149 (2013).

Minor comments

  • In the paragraph on muscle development and regeneration 5.1, the author explains how muscle regenerates in general. Could the author provide information on modifications of this process induced by CKD?

Round 2

Reviewer 3 Report

I would like to congratulate the author for the nice improvements made to the paper since our last reading. The author has fully answered the raised questions i had.